# Heterozygous Arrhythmogenic Cardiomyopathy-*desmoplakin* Mutation Carriers Exhibit a Subclinical Cutaneous Phenotype with Cell Membrane Disruption and Lack of Intercellular Adhesion

**DOI:** 10.3390/jcm10194608

**Published:** 2021-10-08

**Authors:** Eva Cabrera-Borrego, Trinidad Montero-Vilchez, Francisco José Bermúdez-Jiménez, Jesús Tercedor-Sánchez, Luis Tercedor-Sánchez, Manuel Sánchez-Díaz, Rosa Macías-Ruiz, María Molina-Jiménez, Francisco Javier Cañizares-García, Eduardo Fernández-Segura, Angel Fernandez-Flores, Salvador Arias-Santiago, Juan Jiménez-Jáimez

**Affiliations:** 1Cardiology Department, Hospital Universitario Virgen de las Nieves, 18014 Granada, CP, Spain; ecabbor@gmail.com (E.C.-B.); derbifdx@gmail.com (F.J.B.-J.); luis.tercedor.sspa@juntadeandalucia.es (L.T.-S.); rmacias148@msn.com (R.M.-R.); mmolina@fibaosalud.com (M.M.-J.); jimenez.jaimez@gmail.com (J.J.-J.); 2Instituto de Investigación Biosanitaria ibs. GRANADA, Universidad de Granada, 18012 Granada, CP, Spain; tmonterov@correo.ugr.es (T.M.-V.); jesustercedor@gmail.com (J.T.-S.); manolo_94_sanchez@hotmail.com (M.S.-D.); 3Dermatology Department, Hospital Universitario Virgen de las Nieves, 18014 Granada, CP, Spain; 4Centro Nacional de Investigaciones Cardiovasculares Carlos III (CNIC), 28029 Madrid, CP, Spain; 5Department of Histology, University of Granada, 18016 Granada, CP, Spain; fjcg@ugr.es (F.J.C.-G.); efsegura@ugr.es (E.F.-S.); 6Pathological Anatomy Department, Hospital El Bierzo, 24404 León, CP, Spain; dermatopathonline@gmail.com

**Keywords:** arrhythmogenic-cardiomyopathy, desmoplakin, skin-homeostasis, keratinocytes, pseudomolinethrix

## Abstract

Genetic variants that result in truncation in *desmoplakin* (*DSP*) are a known cause of arrhythmogenic cardiomyopathy (AC). In homozygous carriers, the combined involvement of skin and heart muscle is well defined, however, this is not the case in heterozygous carriers. The aim of this work is to describe cutaneous findings and analyze the molecular and ultrastructural cutaneous changes in this group of patients. Four women and eight men with a mean age of 48 ± 14 years were included. Eight met definitive criteria for AC, one was borderline and three were silent carriers. No relevant macroscopic changes in skin and hair were detected. However, significantly lower skin temperature (29.56 vs. 30.97 °C, *p* = 0.036) and higher transepidermal water loss (TEWL) (37.62 vs. 23.95 g m 2 h 1, *p* = 0.028) were observed compared to sex- and age-matched controls. Histopathology of the skin biopsy showed widening of intercellular spaces and acantholysis of keratinocytes in the spinous layer. Immunohistochemistry showed a strongly reduced expression of DSP in all samples. Trichogram showed regular nodules (thickening) compatible with pseudomonilethrix. Therefore, regardless of cardiac involvement, heterozygous patients with truncation-type variants in *DSP* have lower skin temperature and higher TEWL, constant microscopic skin involvement with specific patterns and pseudomonilethrix in the trichogram.

## 1. Introduction

Arrhythmogenic cardiomyopathy (AC) is an inherited myocardial disease that predisposes to fibrofatty replacement, ventricular aneurysms and sudden cardiac death due to ventricular arrhythmias [1]. This disease is one of the leading causes of sudden cardiac death due to ventricular tachyarrhythmias, especially in people under 35 years of age and in athletes. The first genetic variants associated with this condition were described in genes encoding desmosomes, critical structures involved in adherence regulation and mechanical integrity of tissues, especially in the myocardium and the skin. Familial characterization of this disease as a genetically conditioned disorder of desmosomes was first made by studying patients with Naxos disease, an autosomal recessive cardiocutaneous syndrome caused by mutations in *plakoglobin* and featured by the presence of AC, palmoplantar hyperkeratosis and „woolly” hair [2]. Carvajal disease was later described as another recessive cardiocutaneous syndrome, in this case by mutations in *desmoplakin (DSP)*. However, mutations in desmosomal genes with autosomal dominant inheritance have classically been described to cause an exclusive cardiac phenotype without skin involvement [3]. The published series have shown that heterozygous carriers can develop as severe heart disease as homozygotes, although in heterozygosis the penetrance of the disease is incomplete. The causes of this incomplete penetrance are not yet fully understood. The age at onset of AC is variable with an estimated mean age of 31 years, being rare after the age of 60 years [4]. Gender disparity in the phenotypic presentation of AC has been demonstrated in different studies, highlighting male sex as a risk factor for the development of life-threatening arrhythmias [5].

Furthermore, cutaneous keratinocytes have been shown to express all cardiac isoforms of desmosomal proteins [6,7,8]. These findings support the hypothesis that they may reflect the molecular and histological changes that occur in the myocardium of patients affected by AC. Moreover, cutaneous involvement has recently been observed in a cohort of heterozygous patients with AC and truncating *DSP* variants [9].

The aim of this study is to describe cutaneous findings and analyze the molecular and ultrastructural cutaneous changes in heterozygous *DSP* truncating variant carriers with AC. The finding of specific patterns of cutaneous and hair involvement could potentially identify new patients, even in the absence of macroscopic cardiac involvement.

## 2. Materials and Methods

### 2.1. Patients

Patients with AC and a truncating *DSP* variant were selected from the Inherited Cardiovascular Diseases Unit of our center. Eight of the patients fulfilled current AC Task Force criteria (TFC) for left-sided or biventricular cardiomyopathy [10], one was identified as borderline and three cases were considered silent carriers. A complete cardiological study, including an electrocardiogram (ECG), 24 h-ambulatory ECG, exercise test, echocardiogram and cardiac magnetic resonance (CMR) was performed to assess the cardiac phenotype. Genetic test ruled out any other genetic variants and complex inheritance patterns. Pathogenicity of genetic variants was classified according to the American Academy [11], taking into account frequency in public databases (including the Human Gene Mutation Database, Single Nucleotide Polymorphism Database, NHLBI GO Exome Sequencing Project and ClinVar or in the Exome Aggregation Consortium (ExAC) database), its previous literature description, and several bioinformatic predictions according to its localization and conservation. After identification of *DSP* pathogenic variants in the unrelated index patients, we performed a genetic cascade screening among all available relatives using Sanger DNA sequencing method. Phenocopies such as myocarditis, sarcoidosis or ischemic heart disease were reasonably ruled out.

### 2.2. Dermatological Evaluation and Skin Barrier Function Assessment

Two independent dermatologists evaluated skin alterations and scalp disorders in the patients. Plucked hair shafts were collected to perform a trichogram. Homeostasis parameters related to epidermal barrier function were also measured. Transepidermal water loss (in g·m^−2^·h^−1^, using Tewameter^®^ TM 300), stratum corneum hydration (in arbitrary units (AU), using Corneometer^®^ CM 825), pH (using Skin-pH-Meter^®^ PH 905), erythema and melanin index (in AU, using Mexameter^®^ MX 18) and skin temperature (in °C, using Skin-Thermometer ST 500) were measured by a Multi Probe Adapter (MPA, Courage + Khazaka electronic GmbH, Bilbao, Spain). All variables were measured ten times, using their average for analysis. All these measurements were taken following the same order. All measurements were taken in the same room at a mean room temperature of 23 ± 1 °C and ambient air humidity of 45% (range, 40–50%). All participants underwent an adaptation period of at least 20 min before the measurements were taken. No systemic or topical treatments were allowed 3 hours before the measurements were taken. These variables were measured at three body sites: the right volar forearm, an involved area on the right palm and a non-involved area on the right palm. If the patient did not have skin manifestation, measurements were only taken on the right volar forearm and on the right palm.

Healthy controls gender- and age-matched (±3 years) with patients were also measured on the right volar forearm and on the right palm. These volunteers were people who attended the Dermatology Department for trivial conditions such as melanocytic nevi or seborrheic keratoses and did not suffer from an inflammatory skin disease, such as psoriasis or atopic dermatitis.

### 2.3. Samples Processing

A cutaneous biopsy was taken from non lesional skin from the palm of the hand in all cases. Samples were fixed in formaldehyde 10%, cut, processed and included in paraffin. Then, 2 µm sections were obtained on high adhesion slides. The deparaffination plus antigen retrieval was automatically performed using the PTLink from Agilent Dako. The immunostaining was performed in the AutostainerLink 48 from Dako. The slides were revealed with diaminobenzidine and counterstained with hematoxylin. Sample 12 was not valid for immunohistochemical or hematoxylin-eosin study.

### 2.4. Immunohistochemistry

Skin biopsies were studied with hematoxylin-eosin and subsequently an immunohistochemical study was performed to assess the cutaneous expression of different proteins. Information of the antibodies used, the manufacturers as well as the conditions used during the immunohistochemical procedure is included in the Table A1 of Appendix A. Biopsies from our archives from patients without disease (normal skin) were used as controls.

### 2.5. Transmission Electron Microscopy

Skin samples were fixed and embedded according to standard protocols. The tissue sample was fixed by immersion in 2% glutaraldehyde in 0.1 M cacodylate buffer (PB), pH 7.4, for 4 h at 4 °C. After fixation, the samples were rinsed three times with 0.1 M cacodylate buffer (PB), pH 7.4, for 15 min at 4 °C and stored overnight until further processing. Subsequently, the samples were postfixed with 0.1% osmium tetroxide in 0.1 M PB, pH 7.2, containing 1% potassium ferrocyanide for 1 h at 4 °C, dehydrated in a graded series of alcohols and embedded in epoxy resin. Semithin sections (500 nm) were stained with toluidine blue for light microscopic examination. Ultrathin sections were stained with uranyl acetate and lead citrate and analyzed under a Zeiss LEO 906E transmission electron microscope (Zeiss, Oberkochem, Germany).

### 2.6. Statistical Analysis

Descriptive statistics were used to present the sample characteristics. Continuous data was expressed as the mean ± standard deviation. The absolute and relative frequency distributions were estimated for qualitative variables. The Shapiro–Wilk test was used to check the normality of data distribution and Levene’s test to check the homogeneity of variance. The Student’s t-test for independent samples or Welch’s test was used to compare homeostasis parameters between patients and controls. The Student’s t-test for independent samples was used to compare homeostasis parameters between affected and non-affected areas in the same subject. A *p*-value of ≤ 0.05 was considered statistically significant. Statistical Analyses were performed using the SPSS package (SPSS for Windows, Version 24.0 Chicago: SPSS Inc.).

## 3. Results

Twelve heterozygous carriers of genetic variants leading to truncation in *DSP* from seven different families were included—four females (33%) and eight males (66%)—with a mean age of 48 ± 14 years. Eight patients met definitive criteria for the diagnosis of AC according to the 2010 TFC: two with exclusive left ventricular involvement, two with right ventricular involvement and four with biventricular involvement. Most of these cases showed segregation of the *DSP* mutation with a phenotype consisting of left ventricular fibrofatty replacement, low QRS complex voltages and an increased arrhythmic risk (Figure A1 of Appendix B). Patients 4, 6 and 9 had no pathological findings on ECG, echocardiogram or CMR, did not present events during follow-up, and were classified as silent carriers. Finally, one case showed a borderline AC status according to the TFC. A summary of the baseline cardiac phenotype of the enrolled patients is included in Table 1.

### 3.1. Dermatological Evaluation and Skin Barrier Function Assessment

Clinically, only 33.33% (4/12) of patients had dermatological manifestations: 16.67% (2/12) had androgenic alopecia, Ludwig grade I, and 16.67% hyperkeratosis in pressure areas of the hands. Regarding skin barrier function (Table 2), we observed that transepidermal water loss was significantly higher on the patients’ palm than in controls’ (37.62 vs. 23.94 g·m^−2^·h^−1^, *p* = 0.028) while temperature was lower (29.56 vs. 30.97 °C, *p* = 0.036). No differences in stratum corneum hydration, erythema, melanin or pH were found on the palm. Moreover, no differences in skin barrier function on the volar forearm were found between patients and controls. Differences between the involved and the non-involved area on the right palm were not observed in any skin barrier function parameter.

### 3.2. Histopathology

Table 2 shows the main histopathological findings in each patient. The most remarkable feature was a significant widening of the intercellular spaces of the keratinocytes, which was evident in all the cases. This was most noticeable in the spinous layer and, in consequence, the spines of the keratinocytes were more evident than usual. Focally, acantholysis with separation of the keratinocytes from each other and open intercellular clefts (Figure 1A), with evidence of round acantholytic cells (Figure 1B) were observed.

In all the analyzed cases, the keratinocytes showed a very unusual appearance, with a striated cytoplasm which reminded the lines that shape fingerprints. For this reason, the term ‘fingerprint sign’ has been proposed (Figure 2). In addition, papillomatosis, dyskeratosis, transition cells, pale cytoplasm and cytoplasmic eosinophilic globules were absent.

### 3.3. Trichogram

All trichogram findings are summarized in the Table A2 of Appendix B. Some patients showed a prominent shift towards catagen/telogen. There was no hair shaft dysplasia or cuticular defects. However, seven showed non-uniform thickness of hair shafts and pseudomonilethrix (Figure 1C).

### 3.4. Immunohistochemistry

The expression of Plakoglobin, Filamin C and Connexin 43 in the skin biopsies was normal (Table 2). However, most of the patients showed negative results for desmoplakin expression (Figure 1D) and only three of them (1, 5 and 6) showed weak expression (Figure 1E).

### 3.5. Transmission Electron Microscopy

In transmission electron microscopy analysis of skin biopsies, 50% of patients showed that keratinocytes of the spinous layer had a normal structure, characterized by a polygonal/elongated morphology with abundant bundles of tonofilaments (keratin filaments) in the cytoplasm, sometimes associated with masses of keratohyalin (Figure 3A). In these samples, desmosomes showed normal ultrastructural features with the presence of undisturbed electron-dense midline, inner plaques and keratin filament junctions (Figure 3B). For the other 50% of patients, ultrastructural analysis of the spinous layer showed an enlargement of the intercellular space between keratinocytes and an increase in the length of the cytoplasmic bridges (Figure 3C,D), as was found in the histological analysis of the samples. In these cases, a decrease in the number of desmosomes was observed, although they showed an unaffected structure (Figure 3E).

## 4. Discussion

Truncating variants in *DSP* have repeatedly been related with a high penetrance cardiac syndrome with left ventricular fibrofatty replacement, predisposition to malignant arrhythmias and sudden cardiac death [12,13,14,15]. The *Carvajal Syndrome* reflects a recessive inheritance pattern with clinical affection of DSP not only at the cardiac level, but also affecting the skin. A major hallmark of the disease pathophysiology is the presence of cell death leading to fibrosis, predominantly at the epicardium. This location has classically made it difficult to obtain pathological samples to observe typical AC changes. The presence of cutaneous involvement when these variants are present in heterozygosis has previously been reported is small studies, but the real prevalence of skin and ultrastructural underlying cellular changes is yet to be described. Our study confirms the existence of a non-clinically relevant cutaneous phenotype, but in the presence of a highly depletion (or even complete absence) of DSP at the cutaneous level. We have been able to describe novel and potentially very specific cutaneous DSP changes that might be useful for the diagnosis and follow up, as they may mirror ultrastructural changes of the heart. These changes may help, as well, in the interpretation of variants of uncertain significance in *DSP* in patients with AC.

Compared to previous publications [9], in our cohort the macroscopic skin findings are not apparent, but the differences observed in skin homeostasis, with lower temperature and greater transepidermal water loss (TEWL), which had not been described until now, are noteworthy. High TEWL values are related with some skin diseases, such as atopic dermatitis or psoriasis [16]. So, our results might be reflecting subclinical skin impairment in AC patients. On the contrary, Maruthappu et al., found a fully penetrant cardio-cutaneous syndrome in heterozygous *DSP* loss of function mutation carriers [9]. These differences may reflect a different spectrum of *DSP* mutations, as their families carried variants affecting both *DSP* isoforms, while ours affected predominantly the cardiac isoform (*DSP I*). It is likely that our population represent in a better way the real-world AC-DSP-related patients, and we have provided evidence of ultrastructural changes in cell-cell adhesion that might have clinical implications in the future.

The histopathological changes evidenced in Carvajal syndrome and Naxos disease are usually described in the literature as epidermolysis with dyskeratosis and epidermal pseudospongiosis, respectively [17,18]. However, it is difficult to find a more accurate description with detailed histopathologic imaging. In fact, several of the most relevant studies on these conditions do not show the cutaneous histopathological findings [19,20,21,22]. Our patients showed negative (or almost negative) results in the immunohistochemical expression of desmoplakin. Since desmoplakin is located in the central part of the desmosomal plaque, it is not surprising that disruption in the protein lead to the observed findings: widening of the interepidermal space and eventual acantholytic foci [23]. A similar decrease in *DSP* expression has been involved in other acantholytic conditions, such as acantholytic squamous cell carcinoma [24], Hailey-Hailey disease [25], Darier’s disease [25], or epidermolysis bullosa [26]. In contrast, desmoplakin is not decreased (even in acantholytic cells) in the case of autoimmune acantholysis [25]. Winik et al. published very interesting results in two children with acantholytic ectodermal dysplasia. Their histopathological findings shown by electron microscopy are very similar to ours, with widening of the intercellular spaces between adjacent keratinocytes [18]. However, in our study those observations are accompanied by an atypical appearance of the keratinocytes with a striated cytoplasm, which we have termed the *fingerprint sign*. All these observations can be explained by the altered function of cellular desmosomes that may be caused by the truncation of *DSP*. It is particularly interesting that other series of heterozygous patients with pathogenic variants in *DSP* do not show a reduction in *DSP* expression in the skin, but only alterations in the distribution of plakoglobin and connexin 43 [9].

On the other hand, a very high prevalence of pseudomonilethrix was observed, which did not correlate with a curly hair phenotype that was highly prevalent in the previously mentioned study [9].

We found no differences in cutaneous and hair involvement between borderline and silent carriers compared to those who met definitive diagnostic criteria for AC. According to Table 2, the three silent carriers and the patient with a borderline diagnosis showed the same features as those with a definitive diagnosis, including a widening of the intercellular spaces of the keratinocytes, a striated appearance of the keratinocyte cytoplasm (‘fingerprint sign’) and a significant reduction of desmoplakin in the immunohistochemical study. Concerning the study of hair samples, one of the silent carriers showed pseudomonilethrix, while the other two did not. The sample from the patient with borderline diagnosis could not be taken.

These findings suggest that the cutaneous–pilose involvement in these patients is independent of the degree of cardiac involvement, which could be particularly helpful in the diagnosis in early stages of the disease or when cardiac tests are inconclusive.

## 5. Conclusions

Regardless of cardiac involvement, heterozygous patients with AC and truncation-type variants in *DSP* present lower skin temperature and higher transepidermal water loss, constant microscopic cutaneous findings and hair shaft involvement (pseudomonilethrix) with specifical patterns such as the “fingerprint sign”, in the absence of relevant clinically cutaneous findings. Although studies with larger cohorts are needed, as there are notable differences from the findings published before, these results open the possibility to include the skin and hair study as a marker of the disease even in the absence of cardiocutaneous affection.

## Figures and Tables

**Figure 1 jcm-10-04608-f001:**
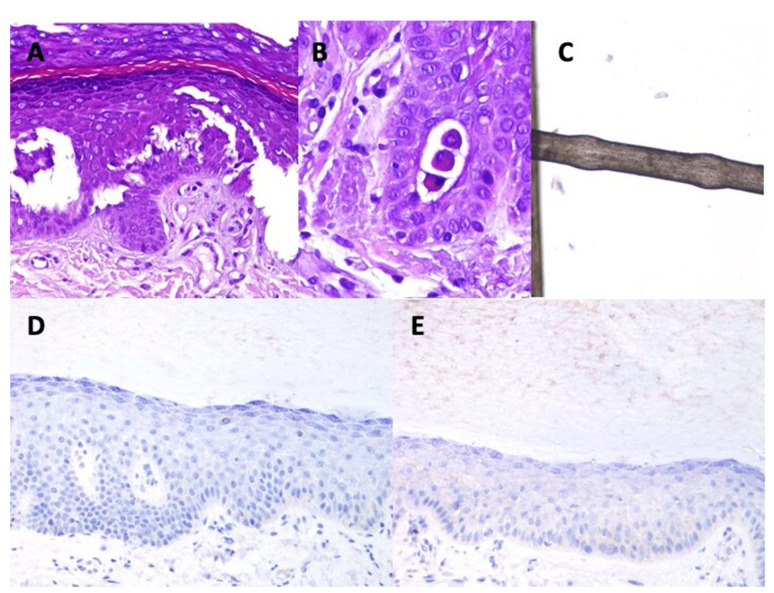
Main findings in the immunohistochemistry study. (**A**): acantholytic areas evidenced in case 9 (Hematoxylin-Eosin ×200). (**B**): Acantholytic round cells. Some desmosomes (spines) are still observed between the round cells and adjacent keratinocytes (hematocillin-eosin ×400). (**C):** monilethrix. (**D**): negative expression of desmoplakin in case 4 (×200). (**E**): weak expression of desmoplakin in case 6 (×200).

**Figure 2 jcm-10-04608-f002:**
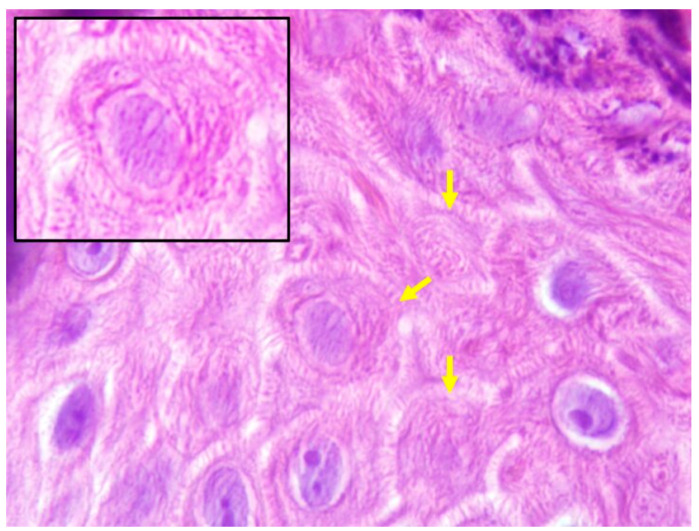
The fingerprint sign. Keratinocytes with striated cytoplasm (Hematoxylin-Eosin ×1000).

**Figure 3 jcm-10-04608-f003:**
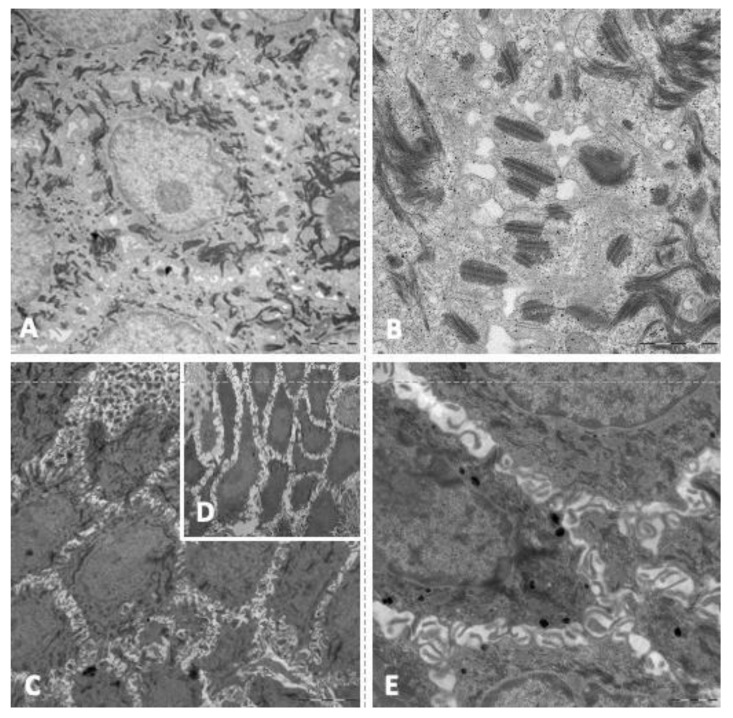
Main findings in the electron microscope study. (**A**): keratinocytes of the spinous layer with abundant keratin filaments in cytoplasm. (**B**): keratinocytes with normal ultrastructural features (electron-dense midline, inner plaques and keratin filament junctions). (**C**): enlargement of the intercellular space between keratinocytes. (**D**): increase in the length of the cytoplasmic bridges. (**E**): Keratinocytes with unaffected structure.

**Table 1 jcm-10-04608-t001:** Baseline cardiac features.

	Age (Years)	Sex	Genetic Variant	LVEF (%)	Status TFC	Late Gadolinium Enhancement	VPB 24 h-ambulatory ECG	NSVT	Sustained Ventricular Arrhythmias
Patient 1	24	Female	p.Asp960Glnfs*16	55	Borderline	Subepicardial patching in LV	102	No	No
Patient 2	28	Male	p.Arg2284*	50	Definitive	Subepicardial circumferential in LV and RV free wall	2916	No	No
Patient 3	61	Female	g.7568140G > A	53	Definitive	Subepicardial circumferential in LV	679	No	No
Patient 4	60	Male	p.Arg2284*	58	Silent carrier	No	21	No	No
Patient 5	48	Male	p.Arg1045*	48	Definitive	Lineal infero-lateral in LV and RV free wall	1544	No	Yes
Patient 6	50	Male	p.Arg1045*	60	Silent carrier	No	51	No	No
Patient 7	39	Male	p.Arg2284*	41	Definitive	Subepicardial circumferential in LV	381	No	No
Patient 8	58	Male	p.Asp960Glnfs*16	30	Definitive	Subepicardial circumferential in LV	3363	Yes	No
Patient 9	31	Male	p.Asp960Glnfs*16	55	Silent carrier	No	4	No	No
Patient 10	62	Female	p.Asp960Glnfs*16	57	Definitive	No	474	No	No
Patient 11	62	Male	p.Ile228Hisfs*29	22	Definitive	CMR could not be performed	29234	No	Yes
Patient 12	53	Female	p.Val2567Cys.Fs*	60	Definitive	No	<500	No	No

LVEF: left ventricle ejection fraction. TFC: Task Force criteria. VPB: ventricular premature beats. ECG: electrocardiogram. NSVT: non-sustained ventricular tachycardia. SVA: sustained ventricular arrhythmia (sustained ventricular tachycardia, ventricular fibrillation, cardiorespiratory arrest or appropriate shock). LV: left ventricle. RV: right ventricle. CMR: cardiac magnetic resonance.

**Table 2 jcm-10-04608-t002:** Main histopathological findings.

	Histopathological findings (H&E)							Immunohistochemistry		
P	Epidermolysis	Parakeratosis	Hiper-Granulosis	Intercellular Spaces	Dehiscence	Fingerprint-Sign	Plakoglobin	Filamin C	Desmoplakin	Conexin 43
1	No	Focal	Yes	Yes	Yes	Yes	Positive	Positive	Positive weak	Positive
2	No	Focal	Yes	Yes	Yes	Yes	Positive	Positive	Negative	Positive
3	No	No	Yes	Yes	No	Yes	Positive	Positive	Negative	Positive
4	No	Focal	Yes	Yes	No	Yes	Positive	Positive	Negative	Positive
5	No	No	Yes	Yes	Yes	Yes	Positive	Positive	Positive weak	Positive
6	No	No	Yes	Yes	No	Yes	Positive	Positive	Positive weak	Positive
7	Yes	No	Yes	Yes	Yes	Yes	Positive	Positive	Negative	Positive
8	Yes	No	Yes	Yes	Yes	Yes	Positive	Positive	Negative	Positive
9	Yes	No	Yes	Yes	Yes	Yes	Positive	Positive	Negative	Positive
10	No	No	Yes	Yes	Yes	Yes	Positive	Positive	Negative	Positive
11	Yes	No	Yes	Yes	Yes	Yes	Positive	Positive	Negative	Positive
12 *	-	-	-	-	-	-	-	-	-	-

P: patient. * Sample 12 was not valid for immunohistochemical or hematoxylin-eosin study.

## Data Availability

Datasets related to this article can be found at https://data.mendeley.com/datasets/3mw76vz26s/1, an open-source online data repository hosted at Mendeley Data (30 July 2021). This article would be part of the PhD thesis research line of Eva Cabrera-Borrego.

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
