# Peer review of "Heterozygous Arrhythmogenic Cardiomyopathy-desmoplakin Mutation Carriers Exhibit a Subclinical Cutaneous Phenotype with Cell Membrane Disruption and Lack of Intercellular Adhesion"

_jcm, 2021, doi:10.3390/jcm10194608_

Round 1
Reviewer 1 Report
This very nicely written study suggests that changes in the skin may provide a window to events occuring in the heart, which is less accessible, and which may porvide a means to better identify afflicted patients. The study is well-designed and presented, and provides an advancement to the field. I have only a few comments to further improve this manuscript.
Major comments
How did cutaneous markers differ between silent carriers and those with overt cardiac manifestations? Is there aa correlation between desmoplakin content and disease burden?
All 3 silent carriers were male. Is this a coincidence or are females and males affected differently in terms of when the disease manifests or to what extent?
The borderline female patient is younger (24) vs. the other two afflicted females (>50). Is this typical of this disease, that the adverse changes develop with age and the borderline patient witll in a few months/years develop definite symptomes? What is the tempral nature of disease manifestation, when does it typically show cardiac sequelae, when cutaneous? The observation that heterozygous carriers lack overt skin changes, is this because of a certain threshold of genetic burden is required for manifestation, or is it a time-question? Basically I am asking: does a heterozygous vs. homozygous carrier, which essentially means a dilution of the burden, develop cardiac and skin manifestations LATER, or do these adverse NOT OCCUR even with delay, because the gene burden is below a certain threshold? Please discuss in furtehr detail, this is a very interesting aspect for lay readers who are not experts in this area.
Minor corrections
P4 line 152 correct <0.05 to P<=0.05
Author Response
This very nicely written study suggests that changes in the skin may provide a window to events occurring in the heart, which is less accessible, and which may provide a means to better identify afflicted patients. The study is well-designed and presented, and provides an advancement to the field. I have only a few comments to further improve this manuscript.
Answer: Dear reviewer, we deeply appreciate your efforts and thoughtful comments that have contributed to improve the manuscript. We sincerely hope that all your comments have been addressed as expected in the revised manuscript.
Major comments:
Comment 1. How did cutaneous markers differ between silent carriers and those with overt cardiac manifestations? Is there a correlation between desmoplakin content and disease burden?
Answer: Dear reviewer, thank you for your observation.
We did not observe significant differences in cutaneous and hair involvement between borderline and silent carriers (patients 1, 4, 6, 9) compared to those who met definitive diagnostic criteria for arrhythmogenic cardiomyopathy (AC).
As described in table 2, the three silent carriers and the patient with a bordeline diagnosis showed the same features as those with a definitive diagnosis, including a widening of the intercellular spaces of the keratinocytes, a striated appearance of the keratinocyte cytoplasm (‘fingerprint sign’) and a significant reduction of desmoplakin in the immunohistochemical study.
Unfortunately, due to the small number of patients recruited, we could not stablish a correlation between desmoplakin content and disease burden. We have expanded the discussion with this information.
Comment 2. All 3 silent carriers were male. Is this a coincidence or are females and males affected differently in terms of when the disease manifests or to what extent?
Answer: Dear reviewer, we believe this is a coincidence mainly due to the sample size. Actually, male gender has been largely reported to be associated with a more malignant course of the disease. We mentioned this aspect of the disease in the introduction referring the excellent work of Mazzanti et al. (JACC, 2016), highlighting male sex as a risk factor in AC.
Comment 3. The borderline female patient is younger (24) vs. the other two afflicted females (>50). Is this typical of this disease, that the adverse changes develop with age and the borderline patient will in a few months/years develop definite symptoms? What is the temporal nature of disease manifestation, when does it typically show cardiac sequelae, when cutaneous? The observation that heterozygous carriers lack overt skin changes, is this because of a certain threshold of genetic burden is required for manifestation, or is it a time-question? Basically, I am asking: does a heterozygous vs. homozygous carrier, which essentially means a dilution of the burden, develop cardiac and skin manifestations LATER, or do these adverse NOT OCCUR even with delay, because the gene burden is below a certain threshold? Please discuss in further detail, this is a very interesting aspect for lay readers who are not experts in this area.
Answer: Dear reviewer, thank you for your comment.
This observation is normal for this condition as a progressive development or worsening of the disease over the years is to be expected.
For heterozygous carriers, AC is a progressive disease characterized by an incomplete penetrance and variable severity of phenotypic expression due to multiple factors (i.e. genetic background, physical activity, or sex). The typical onset of the disease occurs during adolescence, with a mean age at diagnosis in the early 30s. Many patients with AC remain clinically silent and asymptomatic for decades, making the disease difficult to recognize. Since sudden death may occur in the absence of manifest cardiac impairment, our work provides valuable information for early diagnosis.
On the other hand, homozygous patients usually present with have woolly hair in infancy and develop diffuse palmoplantar keratosis during early childhood. In this particular group of patients, the cardiac phenotype is fully penetrant in adolescence with symptomatic arrhythmias and cardiac structural alterations. On the contrary, at this age, a minority of heterozygotes have subtle ECG and echocardiographic changes, but clinically significant disease is not usually observed.
As you suggest, we have included some aspects on the temporal evolution and genetic impact (heterozygous vs homozygous) of this disease based on the review by Smith et al, reference 4, in order to make the manuscript more accessible to readers who are not experts in inherited cardiac diseases.
Minor corrections:
Comment 4. P4 line 152 correct <0.05 to P<=0.05.
Answer: thank you, it has been corrected.
Reviewer 2 Report
The authors described cutaneous findings and analyse the molecular and ultrastructural cutaneous changes in heterozygous DSP truncating variant carriers with AC. They showed specific patterns of cutaneous and hair involvement could potentially identify new patients, even in the absence of macroscopic cardiac involvement.
It seems significant and valuable for readers.
The patients were included borderline and silent carrier patients according to TSC. It means they were subclinical state.
For that reason, please compare cutaneous function of skin biopsy and hair shaft between definitive patients and borderline/silent carrier patients, and discuss it.
Author Response
The authors described cutaneous findings and analyse the molecular and ultrastructural cutaneous changes in heterozygous DSP truncating variant carriers with AC. They showed specific patterns of cutaneous and hair involvement could potentially identify new patients, even in the absence of macroscopic cardiac involvement. It seems significant and valuable for readers.
Dear reviewer, we very much appreciate your effort and thorough analysis of our manuscript. Your comments are of great value and certainly help us to improve our work.
As you have mentioned in your general comment, this work presents a series of cutaneous and hair disorders which have previously not been described in this type of patients, which is of great interest in the evaluation of patients with new genetic variants, of unknown significance, in desmoplakin.
Comment 1. The patients were included borderline and silent carrier patients according to TSC. It means they were subclinical state. For that reason, please compare cutaneous function of skin biopsy and hair shaft between definitive patients and borderline/silent carrier patients, and discuss it.
Answer: Dear reviewer, thank you for your comment. The cardiac, cutaneous and hair findings in each of the patients are described in tables 1, 2 and B1.
In our work, we did not find differences between silent carriers and patients with a definite diagnosis of AC. All patients showed the same features, including a widening of the intercellular spaces of the keratinocytes, a striated appearance of the keratinocyte cytoplasm (fingerprint sign) and a significant reduction of desmoplakin in the immunohistochemical study. We have not performed a statistical analysis of these observations between silent/borderline carriers and those with a definitive diagnosis of AC, due to the small number of patients included.
These findings suggest that the cutaneous-pilose involvement in these patients is independent of the degree of cardiac involvement, which could be particularly helpful in the diagnosis in early stages of the disease or when cardiac tests are inconclusive. Therefore, samples including a larger number of patients are needed to be able to make this assertion.
We have addressed this aspect in the discussion.
We hope we have resolved your concerns with this letter.
We would be pleased to provide you with further information if you deem it necessary.